Achilles tendon thickness and serum asprosin level significantly increases in patients with polycystic ovary syndrome

Ozturk Huseyin Ali drozturkhuseyinali@gmail.com
Arici Fatih Necip
Department of Internal Medicine, University of Health Sciences–Adana Health Practice and Research Center , Adana , Turkey
Foti Daniela
Electronic publication date: 2024 Aug 22
Publication date: 2024
Volume: 12
Electronic Location ID: e17905
Received 2024 Feb 27; Accepted 2024 Jul 22
Copyright: © 2024 Ozturk and Arici
Copyright year: 2024
Copyright holder: Ozturk and Arici
License: This is an open access article distributed under the terms of the Creative Commons Attribution License, which permits unrestricted use, distribution, reproduction and adaptation in any medium and for any purpose provided that it is properly attributed. For attribution, the original author(s), title, publication source (PeerJ) and either DOI or URL of the article must be cited.
License URL: https://creativecommons.org/licenses/by/4.0/

Keywords: Achilles tendon thickness, Asprosin level, Cardiovascular disease, Metabolic disorder, Polycystic ovary syndrome

Funding: The authors received no funding for this work.

==============================
Aim

In our study, we aimed to investigate the Achilles tendon thickness (ATT) and asprosin levels in patients with polycystic ovary syndrome (PCOS) and to evaluate the relationship of these parameters, which may be related to cardio-metabolic diseases.

Methods

In our prospective cross-sectional study, 45 female patients with PCOS and 30 female healthy individuals similar in age were included. Serum dehydroepiandrosterone sulfate (DHEAS), total testosterone, anti-Müllerian hormone (AMH) and asprosin levels were measured using appropriate kits and homeostatic model assessment of insulin resistance (HOMA-IR), luteinizing hormone (LH) to follicle-stimulating hormone (FSH) ratio was calculated. ATT measurements were performed by two radiologists using a high-resolution ultrasound doppler system.

Results

Serum DHEAS, total testosterone, AMH and asprosin levels, HOMA-IR value, LF/FSH ratio, and ATT values were higher in patients with PCOS compared to healthy controls. Correlation analysis was performed between ATT and other parameters in patients with PCOS. In univariate analysis, parameters associated with ATT were detected as asprosin, DHEAS and AMH. In the linear regression analysis performed with significant parameters, asprosin and DHEAS levels were found to be associated with ATT.

Conclusion

ATT values and serum asprosin levels were found to be significantly increased in patients with PCOS, and there is a very close positive relationship between ATT and serum asprosin levels. For this reason, it was thought that ATT measurement could be a cheap, simple and non-invasive monitoring parameter that can be used in the routine cardiometabolic follow-up of patients with PCOS.

Introduction

Polycystic ovary syndrome (PCOS); It is the most common endocrine and metabolic disorder in this age group, affecting 5% to 10% of women of reproductive age (Abacioglu et al., 2021). In patients with PCOS, there are findings related to hyperandrogenism, oligomenorrhea or amenorrhea, hirsutism, and polycystic appearance in the ovaries (Gulumsek et al., 2020). In patients with PCOS, it is possible to find an increase in dehydroepiandrosterone sulfate (DHEAS), luteinizing hormone (LH) follicle stimulating hormone (FSH) ratio, total testosterone, homeostatic model assessment of insulin resistance (HOMA-IR) and anti-Müllerian hormone (AMH) levels (Sumbul et al., 2022). As a result of hormonal and metabolic changes, insulin resistance, diabetes mellitus, obstructive sleep apnea, metabolic syndrome and cardiovascular morbidities may occur. For this reason, early diagnosis and treatment in patients with PCOS is very important to prevent cardiovascular morbidity and mortality (Legro et al., 2013).

The Achilles tendon is the strongest, thickest and largest tendon in the body and has very important functions in leg activities (Fujiwara et al., 2022). The Achilles tendon is evaluated for trauma by magnetic resonance imaging and ultrasonography (USG). Apart from trauma, Achilles tendon thickness (ATT) measurement with USG is recommended as a follow-up method in patients with familial hyperlipidemia (Michikura et al., 2017). Studies have been conducted by Fujiwara et al. (2022), Wang et al. (2018) and Koc et al. (2019) using ATT on cardiovascular disease (CVD) and metabolic syndrome risk factors.

It has been stated in studies that dysregulation of adipokines may play a role in the development of PCOS. PCOS patients tend to be overweight or obese, and adipose tissue can secrete various adipokines (Anagnostis, Tarlatzis & Kauffman, 2018). Asprosin is a novel adipokine secreted by white adipose tissue and is associated with metabolic diseases including diabetes, obesity, PCOS, and CVD (Yuan et al., 2020).

Although the mechanism of PCOS is not fully known, insulin resistance and glucose metabolism disorders are among the accused factors. Considering the strong relationship between PCOS and insulin resistance, metabolic syndrome and its components (diabetes, hypertension, dyslipidemia, cardiovascular disease) should be evaluated in all patients with PCOS. Dysregulation of adipokines may play a role in the development of PCOS. The biological effect of asprosin, an adipokine, on the liver has been shown to be dependent on the G protein-coupled receptor, which activates the adenylyl cyclase-PKA-cAMP responsive element binding pathway, leading to glucose production and release. It has been stated that asprosin may have an effective role in regulating insulin resistance and glucose metabolism (Yuan et al., 2020). Additionally, there are studies indicating that asprosin may be a diagnostic and follow-up marker for atherosclerotic CVD and can be used in the follow-up of cardiometabolic diseases (Shabir et al., 2021). The pathogenesis of PCOS generally involves insulin resistance leading to various cardiometabolic abnormalities (e.g., dyslipidemia, hypertension, glucose intolerance, diabetes, and metabolic syndrome), which puts patients at increased CVD risk group (Farrag et al., 2023; Zhu & Wang, 2023). Therefore, asprosin may be an important biomarker in the cardiometabolic process.

To the best our knowledge, there are no studies related to ATT and asprosin levels in patients with PCOS. In our study, we aimed to investigate whether ATT and asprosin levels in PCOS patients differ from the normal population and to evaluate the relationship between these parameters, which may be related to cardio-metabolic diseases.

Material and method

Study population

In our prospective cross-sectional study, 45 female patients with PCOS Phenotype 1 (hyperandrogenism + oligo-anovulation + polycystic ovarian morphology) and 30 female healthy individuals similar in age were included. PCOS was diagnosed according to the 2003 Rotterdam criteria. According to these criteria, it is recommended to diagnose PCOS if at least two of the following three features are present: chronic oligoovulation or anovulation, clinical or biochemical hyperandrogenism, and polycystic ovaries identified on USG. In patients with chronic renal failure, acute-chronic liver diseases, diabetes mellitus, hypertension, thyroid diseases, rheumatic diseases, malignancies, Achilles tendon trauma, musculoskeletal system disease, immobility problems, peripheral vascular and cerebrovascular disease, valvular heart disease, heart failure, active infections, pregnant women and patients who did not want to participate in the study were excluded from the study. The study was conducted in accordance with the Declaration of Helsinki and was approved by the institutional ethics committee. Written consent forms were explained in detail to all participants and they participated in the study after obtaining written informed consent. A detailed anamnesis and physical examination were performed. Adana City Training and Research Hospital Ethics Committee approved the study with decision number 1652 dated 02.12.2021. The body mass index (BMI) of all patients was measured. Serum glucose, creatinine, sodium, potassium, aspartate aminotransferase (AST), alanine aminotransferase (ALT), calcium, thyroid stimulating hormone (TSH), C-reactive protein (CRP), total testosterone, HOMA-IR, DHEAS, LH, FSH, high-density lipoprotein cholesterol (HDL), low-density lipoprotein cholesterol (LDL), and triglycerides were measured using automated laboratory methods (Abbott Aeroset, Minneapolis, MN, USA) and appropriate commercial kits (Abbott). The HOMA-IR value was calculated by multiplying the fasting insulin level with the blood sugar taken after 8–10 h of fasting and dividing this result by 405.

To measure serum AMH and asprosin levels, blood samples were taken from the patient and control groups at 08:00 in the morning after a 12-h fast and were centrifuged at 4,000 rpm for at least 10 min. Serum samples obtained after centrifugation were stored at −80 °C until the study was performed. To measure serum AMH levels (Beckman Coulter, Inc, Brea, CA, USA), an automated Enzyme-linked Immunosorbent Assay (ELISA) reader (Thermo Fisher Scientific, Vantaa, Finland) and computer program (Scanlt for Multiscan FC 2.5.1) were used. Serum asprosin levels were studied using the ELISA kit from the Cloud-Crone Corporation (catalog no. SEA332Hu). The working range of the kit was 0.156–10 ng/mL (Intra-Assay: CV < 10%, Inter-Assay: CV < 12%).

Achilles tendon ultrasonography

AT USG examinations were performed by two 10 years experienced radiologists in this field using a high-resolution ultrasound Doppler system (Philips EPIQ 7) equipped with a high-resolution linear transducer (12-5 MHz) (Philips Health Care, Bothell, WA, USA). AT B-mode USG evaluation was performed with the patient lying in the prone position, the ankle in approximately 90° plantar flexion and the Achilles tendons at rest. ATT was calculated based on the maximum distance between the anterior and posterior walls, and the Achilles tendon width was measured as the distance from medial to lateral in the transverse plane at the level of the medial malleolus. The penetration depth was set to 2.5 and 4 cm, respectively, to be useful for each patient for all B-mode USG examinations. Subjects were evaluated independently by two experienced radiologists.

Statistical analysis

All analyzes were performed using the statistical software package SPSS 22.0 (Chicago, IL, USA). The distribution of continuous variables was evaluated with Kolmogorov-Smirnov. Within-group continuous variable data are expressed as mean ± SD. Normally distributed continuous variables were compared using Student’s t test, while the Mann-Whitney U test was used to compare differences between two independent groups when not normally distributed. Categorical variables were compared with the chi-square test. The κ coefficient was used to examine the interobserver and intraobserver variability of USG measurements. Parameters associated with ATT were determined by univariate Spearman or Pearson correlation analyses. In multivariate linear regression analysis, the parameters with the closest relationship with ATT were determined. Statistical significance was accepted if p < 0.05.

Results

Cohen’s κ values assessing interobserver variability were above 90% for all Achilles tendon USG measurements (p < 0.001 for all comparisons). The study population was divided into two groups: patients with PCOS and healthy controls. Demographic, clinical and laboratory findings of the study groups were compared. BMI, DHEAS, LH/FSH ratio, LH, FSH, total testosterone, HOMA-IR, AMH, and serum asprosin levels were higher in patients with PCOS compared to healthy controls. When the study groups were compared in terms of Achilles tendon USG parameters, ATT values were higher in patients with PCOS (Table 1, Fig. 1). Correlation analysis was performed between ATT and other demographic, clinical and laboratory parameters in patients with PCOS. In univariate analysis, parameters associated with ATT were asprosin, DHEAS and AMH. Linear regression analysis was performed for parameters that were significantly associated with ATT in univariate analysis. As a result of this analysis, it was determined that asprosin (Fig. 2) and DHEAS (Fig. 3) levels were related to ATT (Table 2). In the correlation analysis performed in the healthy control group, no correlation was found between ATT and asprosin (Table 3).

Table 1 Demographic, clinical, laboratory findings, AMH, asprosin levels and Achilles tendon measurements of patients with PCOS and healthy control group.

Variables	Healthy control group n = 30	Patient with PCOS group n = 45	p	
Age (year)	28.5 ± 4.75	29.2 ± 4.86	0.540	
Body mass index (kg/m2)	23.3 ± 1.66	27.0 ± 7.42	0.002	
Basal heart rate (pulse/minute)	76.7 ± 7.47	79.3 ± 10.4	0.254	
Systolic blood pressure (mmHg)	116.0 ± 4.51	117.5 ± 6.82	0.304	
Diastolic blood pressure (mmHg)	78.6 ± 3.56	80.0 ± 8.66	0.352	
White blood cell (10³/µL)	6.81 ± 1.64	6.21 ± 1.48	0.113	
Hemoglobin (g/dL)	12.8 ± 0.62	12.9 ± 1.60	0.715	
Platelet (10³/µL)	289.2 ± 35.4	278.0 ± 70.3	0.377	
Glucose (mg/dL)	80.0 ± 7.10	81.8 ± 9.02	0.339	
Creatinine (mg/dL)	0.61 ± 0.14	0.59 ± 0.08	0.462	
Sodium (mmol/L)	139.0 ± 2.11	138.9 ± 1.15	0.870	
Potassium (mmol/L)	4.37 ± 0.37	4.38 ± 0.30	0.933	
Calcium (mg/dL)	9.52 ± 0.53	9.67 ± 0.34	0.190	
Aspartate aminotransferase (u/L)	18.2 ± 6.31	20.0 ± 7.80	0.307	
Alanine aminotransferase (u/L)	15.5 ± 4.56	18.6 ± 9.27	0.059	
Triglyceride (mg/dL)	109.1 ± 41.0	121.5 ± 70.0	0.387	
High-density lipoprotein (mg/dL)	53.0 ± 12.3	54.2 ± 14.1	0.692	
Low-density lipoprotein (mg/dL)	120.2 ± 21.7	125.4 ± 37.3	0.499	
Thyroid stimulating hormone (mIU/L)	1.84 ± 0.77	2.04 ± 1.06	0.388	
C-reactive protein (mg/L)	0.19 ± 0.10	0.20 ± 0.15	0.724	
LH/FSH ratio	0.66 ± 0.21	1.55 ± 0.74	<0.001	
LH (IU/L)	4.44 ± 1.28	12.9 ± 6.48	<0.001	
FSH (IU/L)	7.15 ± 2.16	8.42 ± 1.46	0.007	
Dehydroepiandrosterone sulfate (IU/mL)	68.2 ± 14.4	336.3 ± 203.4	<0.001	
Total testosterone (pg/mL)	56.5 ± 9.79	84.0 ± 21.8	<0.001	
Anti-Mullerian Hormone (ng/mL)	1.82 ± 0.49	6.21 ± 4.17	<0.001	
HOMA-IR	1.65 ± 0.42	2.51 ± 0.52	<0.001	
Asprosin (ng/mL)	46.2 ± 4.5	52.6 ± 9.5	<0.001	
Achilles tendon thickness (mm)	4.71 ± 0.54	5.01 ± 0.64	0.037	
Achilles tendon width (mm)	13.7 ± 1.56	14.1 ± 1.63	0.345	
Note:

LH: Luteinizing Hormone, FSH: Follicule Stimulating Hormone, HOMA-IR: Homeostatic Model Assessment Insulin Resistance, PCOS: Polycystic ovary syndrome.

Figure 1 Achilles tendon thickness measurements of patient with polycystic ovary syndrome (A) and healthy control (B).

(A) Achilles tendon thickness measurements of patient with polycystic ovary syndrome. (B) Achilles tendon thickness measurements of healthy control.

Figure 2 Scatter plot diagram between Achilles tendon thickness and serum asprosin level.

Figure 3 Scatter plot diagram between Achilles tendon thickness and serum dehydroepiandrosterone sulfate level.

Table 2 Parameters associated with Achilles tendon thickness in patients with PCOS.

Variables	Univariate analysis	Multivariate analysis	
p	r	p	β	
Asprosin (ng/mL)	<0.001	0.568	<0.001	0.046	
Dehydroepiandrosterone sulfate (IU/mL)	0.002	0.412	0.003	0.001	
Anti-Mullerian Hormone (ng/mL)	0.004	0.390	0.265	0.031	
Notes:

R2 Adjusted = 0.472.

PCOS: Polycystic ovary syndrome.

Table 3 Parameter associated with Achilles tendon thickness in healthy control group.

Variables	Univariate analysis	
p	r	
Asprosin (ng/mL)	0.821	−0.043	

Discussion

The main finding of our study is that ATT value and serum asprosin level are significantly increased in patients with PCOS. Another important finding is increased ATT value in patients with PCOS is closely related to serum asprosin and DHEAS levels. Asprosin is associated with metabolic diseases such as diabetes, obesity, metabolic syndrome, dyslipidemia and PCOS. As it is known, cardiovascular disease risk factors increase in these diseases. Many studies have been conducted on cardiovascular disease and metabolic syndrome risk factors and ATT, and it has been reported that increased ATT may be associated with cardiometabolic diseases. Since we found in our study that ATT values and serum asprosin levels were increased and interrelated in PCOS patients, we think that ATT measurement may be a parameter that we can use in the cardiometabolic follow-up of patients with PCOS.

PCOS is the most common endocrine and metabolic disorder in women of reproductive age. Despite its high prevalence, it is diagnosed late and there is no curative treatment method yet (Melin et al., 2024). Insulin resistance and hyperandrogenism per se or combined put PCOS patients a high-risk group for obesity, dyslipidemia, insulin resistance, type 2 diabetes mellitus, metabolic syndrome and cardiovascular diseases in the future. Therefore, early diagnosis and treatment is important (Palomba et al., 2015).

Studies on ATT are generally on hypercholesterolemia, and xanthomas are seen in the Achilles tendon in patients with hypercholesterolemia. The earliest symptom of Achilles tendon xanthoma is thickening of the tendon. Studies have shown that ATT may be a risk factor for CVD in patients with familial hypercholesterolemia (Sugisawa et al., 2012). Studies have stated that it may also be a determinant of atherosclerosis and should be evaluated in individuals with high CVD risk (Genkel et al., 2020). To the best our knowledge, there are no studies evaluating ATT in patients with PCOS. In our study, although the groups were similar in terms of lipid profile, we found the ATT value to be significantly higher in patients with PCOS compared to the healthy control group.

It has been stated that due to the strong correlation between serum AMH levels, which is a determinant of insulin resistance, and the number of follicles on ultrasound, AMH can be used in the diagnosis as an alternative marker of ovarian dysfunction in PCOS (Rudnicka et al., 2021). Studies have shown that serum DHEAS levels are higher in patients with PCOS than in normal individuals and are important in the evaluation of hyperandrogenism (Khan et al., 2021). In our study, we found that AMH and DHEAS levels, which are thought to be important in diagnosing and monitoring PCOS, were higher in the PCOS group and were associated with ATT in the correlation analysis. In the linear regression analysis, we found that there was an independent relationship between ATT and DHEAS levels. In our study; since ATT is found to be related to AMH and DHEAS, which are important laboratory values in the diagnosis and follow-up of patients with PCOS, we think that ATT levels can be measured in the diagnosis and follow-up of patients with PCOS and can be used to predict the activation and metabolic effects of the disease.

Recent studies have shown that asprosin plays an important role in metabolic diseases. Asprosin, which has been associated with metabolic diseases including diabetes, obesity, PCOS, and CVD, is a hormone produced from white adipose tissue (Li et al., 2018). PCOS is a disorder associated with insulin resistance, obesity, cardiovascular risk factors and metabolic disorders. In a study conducted by Li et al. (2018) serum asprosin levels in patients with PCOS were found to be significantly higher than in healthy controls, and it was stated that serum asprosin level could be a potential predictive factor for women with metabolic-related diseases. In the study conducted by Alan et al. (2019) it was found that circulating serum asprosin levels in women with PCOS were higher than in controls, and this was associated with insulin resistance. In the study conducted by Zhu & Wang (2023) high serum asprosin levels were detected in patients with PCOS and it was reported that it could be used as a new indicator in the evaluation of insulin resistance. In our study, we found that serum asprosin level and HOMA-IR score were higher in the PCOS group. In our study, Despite excluding diabetes and CVD patients from our study, our groups showed similarity in terms of other additional metabolic risk factors than PCOS and so, we deduce that high levels of serum asprosin in the PCOS group is very pertinent factor.

In our study, the independent correlation between ATT and serum asprosin level, which is associated with insulin resistance and cardiometabolic diseases, in patients with PCOS suggested that these two parameters could be used in monitoring the cardiometabolic process of this disease. More studies are needed to determine whether asprosin can be used as a biomarker to predict and diagnose PCOS.

Our study has some important limitations. Our study consists of newly diagnosed patients with PCOS. If this study had been conducted in a group of patients previously diagnosed with PCOS and followed up, the diagnostic utility of ATT and asprosin level could have been better evaluated. In our study, patients were not followed. Follow-up studies including other PCOS phenotypes are needed in the same patient group.

Conclusion

As a result, ATT values and serum asprosin levels were found to be significantly increased in patients with PCOS, and there was a very close positive relationship between ATT and asprosin levels. For this reason, it was thought that ATT measurement could be a cheap, simple and non-invasive monitoring parameter that could be used in the routine cardiometabolic follow-up of patients with PCOS.

Supplemental Information

Supplemental Information 1 Patients with PCOS dataset.

Supplemental Information 2 STROBE checklist.

I would like to thank radiology specialist Dr. Burcak Cakir Pekoz for her contribution.

Additional Information and Declarations

Competing Interests

Author Contributions

Human Ethics

Data Availability

The authors declare that they have no competing interests.

Huseyin Ali Ozturk conceived and designed the experiments, performed the experiments, analyzed the data, prepared figures and/or tables, authored or reviewed drafts of the article, and approved the final draft.

Fatih Necip Arici conceived and designed the experiments, performed the experiments, analyzed the data, prepared figures and/or tables, authored or reviewed drafts of the article, and approved the final draft.

The following information was supplied relating to ethical approvals (i.e., approving body and any reference numbers):

Adana City Training and Research Hospital Ethics Committee approved the study with decision number 1652 dated 02.12.2021.

The following information was supplied regarding data availability:

Raw data are available in the Supplemental Files.

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
