# Peer review of "Achilles tendon thickness and serum asprosin level significantly increases in patients with polycystic ovary syndrome"

_PeerJ, doi:10.7717/peerj.17905_

## Round 0.1 · original submission · Major Revisions

The reviewers provided valuable comments to be addressed by the authors.

Reviewer 1 ·

Basic reporting

In this article, the authors explored the relationship between ATT values and serum Asprosin levels in PCOS patients.
1.You should revise your English writing carefully to eliminate small mistakes in your paper and make it easier to understand. There are some grammatical mistakes in the article and the expression of some content is not clear enough. The authors are advised to thoroughly review the manuscript and make any necessary revisions. For example, in the background, the punctuation in the first sentence is not standard.
2.The format of the references also did not correspond to the requirements of the journal.
3.The content in The 'STROBE Statement' is partially missing.
4.What is the relationship between Achilles tendon thickness and overall obesity? Is it related to other factors? Such as body fat percentage, body fat distribution, or daliy physical exertion.

Experimental design

No comment.

Validity of the findings

No comment.

Additional comments

No comment.

Reviewer 2 ·

Basic reporting

Dear Authors,

First of all, I would like to congratulate you for investing the time and resources in researching such a pertinent theme. PCOS is currently a condition that has a considerable prevalence among women in fertile age - which includes women from our workforce that suffer with some serious symptoms. PCOS is related with metabolic alterations impacting the cardiovascular health of the individual. Given that metabolic related disorders are one of the heaviest burdens in several Health Systems around the world, especially in developed countries, it urges to find sophisticated and less invase but still reliable alternative diagnostic means for a precocious intervention and control/preservation of the metabolic and cardiovascular health of women struggling with PCOS. To my knowledge so far, I confirm not finding any study trying to relate ATT, asprosin levels and metabolic markers in patients with PCOS. For that, this work is a novelty and a step ahead in the field.
In general, it is possible to say that :
The English is clear and unambiguous throughout the article.
Literature references are mostly recent and in line with the subject of the studied question in the research work. It could be considered for sufficient field background/context provided, though it would be interesting to supply a tad wider back up for evidence referred throughout the article.
The article, itself is constructed in a rather coherent and professional backbone, including figures and tables – for a matter of aesthetics I suggest keeping both article and tables with the same font. Pertinent raw data are shared. It would be interesting, though, to understand the lower and higher values for the gaps found in most relevant biochemical and anthropometric criteria, such as: age, Body mass index , White blood cell (due its relation with the stress hormones such as cortisol that sometimes appear altered in women with PCOS), Dehydroepiandrosterone, Total testosterone, LH/FSH, Anti-Mullerian Hormone, HOMA-IR, Asprosin, AT parameters. Plus, the polar values of Hemoglobin, Glucose, Aspartate aminotransferase , Alanine aminotransferase, lipoproteins, Thyroid stimulating hormone, C-reactive protein. The results for LH and FSH per se,could also be included, although the most valuable criteria, is no doubt, the ratio presented: LH/FSH.
The conclusions are sober, self-contained with relevant results to the initial research question. Moreover, it serves as an ongoing step for the next move in this field.

The abstract is concise, clear and well structured. I would suggest minor corrections, according to my point fo view:

Line 75: “In patients with PCOS, there is an increase in (...) “
Though it might occur it's advisable not too generalize the findings for every PCOS patient; PCOS signs and symptoms can vary considerably from patient to patient and depending on each phenotype displayed by the individual. In that sense, I would recommend to report it as: “In patients with PCOS, it is possible to find an increase in (...)”
Line 78-80: “As a result of hormonal and metabolic changes, insulin resistance, diabetes mellitus, obstructive sleep apnea, metabolic syndrome and cardiovascular morbidities may occur.”
Although there is a reference provided after the following sentence I suggest providing a solid and varied compilation of references regarding each of the issues that might go hand in hand with PCOS condition.
Line 86-87: “Many studies have been conducted using ATT on cardiovascular disease (CVD) and metabolic syndrome risk factors “
For underlining the importance of the research of the present work, I kindly suggest the introduction of the previous authors who successfully proved the possible relation between ATT and cardiovascular disease markers as well justifying the reason of choosing a particular protocol for this experimental work.
Line 90-91: “Asprosin is a novel adipokine secreted by white adipose tissue and is associated with metabolic diseases including diabetes, obesity, PCOS, and CVD (10).
The importance of asprosin for the PCOS condition is stated in the article. I acutally suggest moving the text from Line 214-228 to the follow the paragraph that ends at Line 91 (Introduction). Furthermore, it would be interesting for the authors to give a deeper insight of the physiologic role and its significance as a potential therapeutic target and/or a biomarker of cardio-metabolic disease as well the relation between this adipokine and the metabolic alterations ocurring in the body, especially in PCOS. It would help the reader, not only to better follow the in depth rational of the authors but also to recognize the importance of the recent discovery within the adipokine universe. It is also important to consider eventual incoherent results of studies correlating asprosin levels in PCOS individuals. Example of review that you can base on:
1. https://pubmed.ncbi.nlm.nih.gov/34108103/
2. doi: 10.3389/fendo.2022.1101091
Actually, if we take into account all PCOS phenotypes identified and recognized so far by the medical and scientific community, probably it is possible to find differential correlations between the asprosin marker and phenotypes. So it would be interesting to bring that point into your work which is a very welcome initiative in the field. If possible, with the information that you have from the diagnosis, I suggest the authors to divide the patients within the 4 possible diagnosable phenotypes and check how the several criteria fall within these four cardinal points (https://academic.oup.com/jcem/article/91/3/781/2843261?login=false: helps to relate the Rotterdam criteria that were considered in this work for diagnosis and . It could give actually a great comparison to applied in radar charts with very valuable information to bring to the table in this hot topic. For furher guidance I leave here some articles:

1. https://pubmed.ncbi.nlm.nih.gov/30325247/ (relates IR with asprosin in PCOS women)
2. DOI: 10.1097/AOG.0000000000002698 (main signs and symptoms of each phenotype)
3. https://pubmed.ncbi.nlm.nih.gov/38285626/ (identified the main hormonal changes in PCOS- it could help you to better detail the relation between asprosin, metabolic changes and a trend to inflammation due to specific hormonal changes that can evetually be associated with subinfertility and infertility in women diagnosed with PCOS. It is know that certain phenotypes have a high association to a larger abdominal circumference and hence inflammation, IR and subfertility)
4. The results of some studies refer that the levels of asprosin could be lower in phenotypes associated with hyperandrogenism (phenotype A, B and D according to Rotterdam classification: https://academic.oup.com/jcem/article/91/3/781/2843261?login=false) when compared with non-hyperandrogenic featured women (phenotype C according to Rotterdam criteria: https://academic.oup.com/jcem/article/91/3/781/2843261?login=false). Please consult this article for confirming the information: doi: 10.1590/1806-9282.20201147.

Experimental design

In regards to the experimental design:
Original primary research within
The research question is well defined, relevant and in tune with the relevant latest findings in the field. It is stated how research fills an identified knowledge gap.
Regarding methodology it would be positive to justify why it was not possible to meet the equality in number of participants for both: Control and Study groups. Usually it is harder to find patients officially diagnosed with PCOS (actually because it is a commonly underdiagnosis or misdiagnosed condition). So, in this case why was it more difficult to find healthy women within a certain age range to participate in the experiment. And, how was the calling made to attract the participants.
Line 131: “AT USG examinations were performed by two experienced radiologists”. It would be advisable to clarify the type of experience of the radiologists. Usually by experienced, it is expected at least 5 years of experience in the field
The statistical analysis is well established. I suggest discriminating the kind of study of the variables was applied in this case. Was it fully blinded (the statistical analysts had no contact and information with the professionals who executed the measurements and with any participant for eg?) How were the results handed to the analysts? Was the statistical analysis performed in a computer with no connection to the internet for security reason?

Validity of the findings

Results and interpretation:
Page 17: Please correct pcos to PCOS (capital letters), and achilles to Achilles. Once again, I suggest making sure that the type of font and size match throughout all the text to engage the attention of the reader.
Line 168-169: As suggested before, it would be interesting to discriminate the variance of ATT and asprosin levels among the different PCOS phenotypes found in this study.
Line 170: “wth PCOS” correct to “with PCOS”
Morover, in Line 170, I advice the authors to add information within a justification for the bridge between their thoughts on linkin the results for the ATT and asprosin, DHEAS levels and the importance of keeping in check metabolic syndrome and CVD patients (for examples. It would help the naive readers to follow better the author's rationale.
My humble suggestion for Line 174: “Despite its high prevalence, PCOS is often diagnosed late and an effective treatment to address such a complex constellation of symptoms has not yet been found .”
If possible, i would add a references of other authors confirming this fact. For example:
DOI: 10.5802/crbiol.147
DOI: 10.1024/0300-9831/a000802
DOI: 10.3390/medicina60020244
DOI: 10.1111/cen.14983
Line 175: “Insulin resistance, hyperandrogenemia or the presence of these two puts PCOS patients”. I suggest simplifying the English as in: “Insulin resistance and hyperandrogenism per se or combined put PCOS patients a high (...)”
Line 185: I encourage be taken into consideration the fact that although the average value for ATT was relatively higher for the study group, it is also true that the control group was considerably smaller hence with less prone to variability.
Line 193-195: “In the linear regression analysis, we found that there was an independent relationship between ATT and DHEAS levels.” Regarding this affirmation i recommend to clearly justify how you concluded that ATT are not interdependent with DHEAS for the better understanding of the naive reader. Moreover, I suggest making an homologous consideration regarding the relation between ATT and AMH.
Line 201: “is a hormone produced from white adipose tissue.” Please provide a direct reference.
Line 201: “PCOS; It is a disease associated with”, please correct it to “PCOS is a disorder (...)”.
Line 205-205: “In the study conducted by Alan et al.,” Reference incomplete, please include the year of the study after the comma. The same for the reference in the line 207-208 and
Line 201-213: “In our study, diabetes and proven CVD were exclusion criteria. Our groups were similar in terms of additional metabolic risk factors other than PCOS. For this reason, we think that high serum Asprosin levels in the PCOS group is an important finding.” I suggest a rephrasing: “ Despite excluding diabetes and CVD patients from our study, our groups showed similarity in terms of other additional metabolic risk factors than PCOS and so, we deduce that high levels of serum Asprosin in the PCOS group is very pertinent factor”.
The limitations of the study are well stated and for that I dare the authors to leave a proposal of a next study in the field, considering what could be improved from this one.

Additional comments

Overall considerations:
This study is overall a stepping stone considering how welcoming it would be to find a diagnosis mean that would serve both purposes: avoiding invasing procedures and getting an early detection of metabolic unbalance in PCOS patients. However, a few considerations should be taken before pusblishing. For example, the fact that PCOS is a condition with a very broad constellation of signs and symptoms that feature three main aspects: reproductive, metabolic and mental health. This work incides mainly in metabolic health since and its the parameter that mostly affects the woman during the different stages of her life besides the fertile age and also constitutes the major budget burden for the national health systems through out the world. Giving that, it helps to identify and differentiate the study group breaking it down to the possible 4 phenotypes (A-D) that are possible to find according to the Rotterdam criteria considered for the study. Although the small pool of this prospective cross-sectional study doesn't allow a reliable projection of the distribution of the 4 phenotypes among the national population, it would allow at least to help to understand if there is an relation of a specific phenotype with the metabolic characterization, including the nouvelle adipokine and the ATT parameter. For a better comprehension and for reading simplification I suggested a modification in the position of some information as well as to add some more references to make the contextualization and justification of the study stone proof. The conclusions are sober, well stuited and above all linked to original research question not overestimating the supporting results. The results consider a whole panoply of criteria which is important in such a complex disorder, though they are analysed in a very simple (though not reductive) and straightforward manner. I salute the quality and novelty of the developed work by the group.

---

## Round 0.2 · Minor Revisions

To further improve the quality of the manuscript, please address the remaining issues indicated by reviewer 1.

Reviewer 1 ·

Basic reporting

Thank you for your response to my comments. Upon re-reading the article, I have identified new issues:

The x-axis of Figure 2 and Figure 3 are identical, yet the number of individuals corresponding to each ATT value is different. It appears that some data points are missing in Figure 2, as it does not contain all 45 points. To further investigate this, I carefully examined the attached Excel spreadsheet and found that there are no missing values for asprosin in the PCOS group. This is a significant bug.

Additionally, I noticed an issue regarding the diagnosis of PCOS. According to the Excel spreadsheet, some PCOS patients have very low AMH values, as low as 1.9. Why?Could you please provide the results of ultrasound-based follicle counts for these patients? It would also be helpful to include the FSH and LH values for each patient. Thank you.

It would be beneficial to include a similar graph, like Figure 2, depicting the factors associated with asprosin in the control group as well.

Experimental design

See above

Validity of the findings

See above

Additional comments

See above

Reviewer 2 ·

Basic reporting

'no comment'

Experimental design

No comment.

Validity of the findings

'no comment'

---

## Round 0.3 · Minor Revisions

In addition to LH/FSH ratio, please provide absolute FSH and LH values, as suggested by reviewer 1.

Reviewer 1 ·

Basic reporting

Thank you for the prompt response from the authors. However, I still believe that the authors should provide the specific values of FSH and LH for each individual, rather than solely presenting LH/FSH ratios. Additionally, demonstrating the correlation between asprosin and ATT in the control group is relatively straightforward, and I am uncertain as to why the authors are adamant about not pursuing this analysis. I believe that conducting such an analysis would be beneficial for readers to interpret the significance of the results. Furthermore, if there were similar correlations between asprosin and ATT in both the control and PCOS groups, then perhaps the interpretation of the conclusions should be approached cautiously. In my opinion, the authors should at least include the data in the supplementary materials for transparency.

Experimental design

see above

Validity of the findings

see above

Additional comments

see above

Reviewer 2 ·

Basic reporting

'no comment'

Experimental design

'no comment'

Validity of the findings

'no comment'

---

## Round 0.4 · accepted · Accept

The authors have addressed the remaining issues raised, so that in the present form the manuscript can be accepted for publication.